# Gut–Adipose Tissue Axis and Metabolic Health

**DOI:** 10.3390/cimb47060424

**Published:** 2025-06-06

**Authors:** Sanja Borozan, Cornelius J. Fernandez, Adnan Samee, Joseph M. Pappachan

**Affiliations:** 1Department of Endocrinology, Clinical Centre of Montenegro, Faculty of Medicine, 81000 Podgorica, Montenegro; sanja_radoman@yahoo.com; 2Department of Endocrinology and Metabolism, Pilgrim Hospital, United Lincolnshire Hospitals NHS Trust, Boston PE21 9QS, UK; drcjfernandez@yahoo.com; 3Department of Medicine, Royal Stoke University Hospital, Stoke-on-Trent ST4 6QG, UK; docsamee@gmail.com; 4Faculty of Science, Manchester Metropolitan University, Manchester M15 6BH, UK; 5Department of Endocrinology, Kasturba Medical College, Manipal, Manipal University of Higher Education, Manipal 576104, India

**Keywords:** gut microbiome, adipose tissue, obesity, metabolic health, gut dysbiosis, gut–adipose tissue axis

## Abstract

The gut–adipose tissue axis plays a crucial role in metabolic health. It is a two-way communication pathway between the gastrointestinal tract and adipose tissue. This axis influences physiological processes vital for maintaining metabolic health, including energy homeostasis, lipid metabolism, and inflammation. Emerging research suggests that the gut microbiota, composed of trillions of microorganisms residing in the intestines, significantly impacts this axis by modulating host metabolism. An imbalance in the gut microbiota (dysbiosis) has been linked to obesity, insulin resistance, and other metabolic disorders. Innovative therapeutic strategies and dietary interventions aimed at modulating the gut–adipose tissue axis have shown encouraging results in improving metabolic health. A deeper critical understanding of the gut–adipose tissue axis is, therefore, essential in understanding the pathophysiology of metabolic disorders so that targeted interventions can be developed to prevent and treat these metabolic disorders. This article highlights the need for integrative approaches that consider both gastrointestinal and adipose functions in metabolic health management.

## 1. Introduction

Adipose tissue (AT), which plays a crucial role in maintaining metabolic health, comprises white adipose tissue (WAT) and brown adipose tissue (BAT), both of which secrete distinct sets of cytokines [1]. WAT is made up of adipocytes with a unilocular lipid droplet and a low mitochondrial density, whereas BAT is made up of adipocytes with multilocular lipid droplets and a high mitochondrial density. WAT is involved in energy storage (mainly triglycerides) and the release of cytokines, including leptin, adiponectin, adipsin, omentin, tumor necrosis factor-alpha (TNF-α), interleukin-6 (IL-6), resistin, visfatin, monocyte chemoattractant protein-1 (MCP-1), plasminogen activator inhibitor-1 (PAI-1), and retinol-binding protein 4 (RBP4) [1]. On the other hand, BAT is involved in energy expenditure, thermogenesis, insulin sensitivity, and the release of cytokines, including fibroblast growth factor 21 (FGF21), bone morphogenetic protein 7 (BMP-7), vascular endothelial growth factor A (VEGF-A), irisin, nesfatin-1, chemerin, meteorin-like protein (METRNL), neuregulin 4 (NRG4), IL-6, interleukin-8 (IL-8), and interleukin-10 (IL-10). In the lean state, the production of anti-inflammatory cytokines, including adiponectin, adipsin, omentin, IL-4, and IL-10, increases, and pro-inflammatory cytokines, including leptin, resistin, chemerin, TNF-α, IL-6, and MCP-1, decrease. The opposite happens in an obese state, increasing the odds for metabolic diseases [1].

The gut microbiome (GM) is a complex, dynamic microbial ecosystem that includes various bacteria, fungi, viruses, and protozoa colonizing the gut, primarily in the caecum and proximal colon [2]. These microsomes communicate with each other, exchange nutrients through cross-feeding, undergo genetic recombination, and evolve together [3]. They maintain a symbiotic relationship with the human host and exert physiological functions like metabolism, vitamin synthesis, hematopoiesis, protection against pathogens, and development/maturation of the immune system [3]. The gut microbial composition is influenced to a smaller extent by the host genetic factors but to a larger extent by the host environmental factors, including diet, physical activity, geographical distribution, lifestyle, hygiene, infection, antibiotic use, and mode of delivery [4,5].

The GM exerts a bidirectional/multidirectional relationship with various other organs. This explains the concept of the gut–organ axis, the important examples of which are the gut–brain axis, gut–pancreas–liver axis, gut–kidney–heart axis, gut–bone axis, gut–lung axis, gut–skin axis, and most importantly, gut–adipose tissue axis [6,7,8,9,10,11]. This article aims to discuss the genetic, epigenetic, and molecular aspects of the gut–adipose tissue axis and its modulation to improve metabolic health.

## 2. Gut–Adipose Tissue Axis

In recent years, research on the two-way communication network between the GM and AT has expanded the landscape of metabolic health regulation and significantly improved our understanding of the potential clinical implications of gut–adipose axis modulation. As a highly active metabolic and endocrine organ, AT is marked by the secretion of a broad spectrum of cytokines, adipokines, and other signaling molecules that influence the tissue susceptibility to gut-derived metabolites [1]. Dietary habits are the main variable in shaping the individual GM profile, favoring or suppressing certain microbial types. The ingested substrate is metabolized into a variety of byproducts, including short-chain fatty acids (SCFAs), secondary bile acids (SBAs), trimethylamine N-oxide, and branched-chain amino acids [12].

According to our current knowledge, SCFAs, such as acetate, propionate, and butyrate, which are the biological end-products of microbial fermentation of poorly digestible polysaccharides in the gut, play a pivotal role in modulating the metabolic pathways involved in obesity and type 2 diabetes mellitus (T2DM) [13]. The existing microbial profile determines the proportion of particular SCFAs. Through biochemical conversions of SCFAs, the energy is produced and is subsequently used to maintain the tight junctions and the intestinal mucosal integrity [14]. Furthermore, SCFAs also stimulate the secretion of gut hormones, such as glucagon-like peptide-1 (GLP-1) and peptide YY (PYY), and neurotransmitters, such as gamma-aminobutyric acid (GABA) and 5-hydroxytryptamine (5-HT), mediating several immune and endocrine pathways and modulating both gut–adipose tissue and gut–brain axes [13,15]. In animal studies, GM alters leptin sensitivity and modulates diet-induced hypothalamic inflammation in the brain via GLP-1 receptor (GLP-1R) activation [16].

Butyrate and propionate stimulate the peroxisome proliferator-activated receptor gamma (PPAR-γ) expression, strongly influencing adipocyte differentiation and lipid storage in AT and protecting against local inflammation [17]. Apart from its anti-inflammatory properties, butyrate also inhibits pancreatic insulin secretion, thereby lowering hyperinsulinemia and improving insulin sensitivity [18]. Conversely, imbalances in GM (dysbiosis), both qualitative and/or quantitative alterations, might contribute to alterations in energy release from substrates and chronic low-grade inflammation [19].

A bidirectional relationship in the gut–adipose tissue axis is achieved through the influence of adipokines, dominantly leptin and adiponectin, on GM composition. A hallmark Danish study identified ‘*low gut bacterial richness**’* as a predictive factor for obesity, insulin resistance (IR), dyslipidemia, and progression to metabolic diseases [20]. In concordance with recent evidence suggesting the role of GM in the pathogenesis of obesity, T2DM, and metabolic dysfunction-associated fatty liver disease (MAFLD), many questions have been raised, though the answers are still awaiting elucidation [19].

## 3. Genetic and Epigenetic Aspects of Gut–Adipose Tissue Axis

The burden of the five most common metabolic diseases, including hypertension, obesity, T2DM, hypercholesterolemia, and MAFLD, has increased over the last three decades from 1.6- to 3-fold globally [21]. Adverse lifestyles, various environmental factors, and genetic susceptibility primarily drive their onset. According to our current knowledge, one of the main mediators bridging the external factors and disease outcomes is the gut–AT axis, through which qualitative and quantitative alterations of GM trigger metabolic and epigenetic changes.

Current research estimates that GM comprises 10^13^–10^14^ microbes from thousands of species, and the sum of their genome, sometimes referred to as the metagenome, contains 150 times more genes than the human genome [22]. The composition and function of the GM consortium are highly individual, with notable dynamic variation across the lifespan: from the first bacterial colonization during infancy and the beginning of SCFA production to breastfeeding, nutrition in childhood, antibiotic use, and lifestyle factors until adulthood, from which point the GM remains relatively stable, in the absence of disease [5]. The most preponderant enterotypes are phyla *Firmicutes*, comprising species *lactobacilli*, *Clostridium*, and *Enterococcus*, and phyla *Bacteroidetes*, comprising the species *Bacteroides* [23].

Several studies delineated associations between specific microbe taxa and host genetics [24,25]. Still, the ongoing dilemma is the proportion of inherited vs. acquired factors in shaping the individual GM. Until now, genetic association studies, together with studies on twins and genome-wide association studies (GWAS), have revealed 110 genetic loci in humans associated with the presence of specific gut microbes [22]. The interactions between the host genome and GM are proven to be highly significant in mouse models. Still, it seems that in humans, environmental factors and diet play a far more important role [26].

Epigenetics is an evolving field that analyses the phenotypic modulation of gene expression not caused by altering the DNA sequence. Epigenetic modifications are perceived as a way to integrate diet and environmental factors and translate them into adaptive mechanisms to maintain homeostasis. The number of publications exploring the link between GM and epigenetic programming is rapidly growing, providing new perspectives on metabolic health and disease. Changes in GM or produced metabolites contribute to the initiation and progression of metabolic diseases via the induction of epigenetic processes, mainly DNA methylation, histone modifications, and regulation by noncoding RNAs (ncRNAs) [5,13,27].

Butyrate, one of the SCFAs, acts as a histone deacetylase (HDAC) inhibitor, enhances histone acetylation, induces regulatory T-cell development, and suppresses the activation of nuclear factor-κB (NF-κB), a potent regulator of immune and inflammatory response [23,28]. It also modulates the transcription of multiple genes associated with obesity [29]. Similarly, acetate, another SCFA, could act as an epigenetic metabolite to enhance histone acetylation and promote lipid synthesis [29]. On the other hand, various epigenetic metabolites, including propionate, folate, and choline, could enhance DNA methylation and thereby modulate the transcription of multiple genes associated with obesity [29].

The ncRNAs, such as microRNAs (miRNAs) and long non-coding RNAs (lncRNAs), are RNA transcripts that are not translated into proteins. In the past decade, miRNAs have been extensively studied for their role in regulating gene expression and cellular functioning. However, recent studies have shown that they are also important in the modulation of the GM–AT axis. A reduced miRNA-155 expression was linked to obesity (adipocyte volume and macrophage infiltration in subcutaneous AT), WAT insulin resistance, and T2DM [30]. The miRNAs have a bidirectional relationship with GM. The host miRNA is engaged in shaping GM while GM induces specific miRNAs implicated in metabolic processes, targeting WAT inflammation and intestinal barrier integrity [31]. Polyphenols could act as an epigenetic metabolite that could revert obesity-related miRNA dysregulation, leading to the induction of weight loss [32].

## 4. Role of the Gut Microbiome in Adipose Tissue Expansion

With the constant rise in the proportion of overweight and obese individuals worldwide, obesity-related metabolic diseases such as T2DM, hypertension, cardiovascular disease (CVD), and MAFLD are becoming more serious threats to public health. The corpus of research on GM in the last two decades, based on animal and human interventional studies, opened new perspectives in our understanding of the complex interplay involved in obesity pathogenesis, suggesting possible innovative therapeutic approaches [33,34].

In a randomized controlled trial, Vrieze et al. infused intestinal microbiota from male lean donors into recipients with metabolic syndrome (MetSy) and observed the changes in the recipients’ GM composition and glucose metabolism [35]. After six weeks, men with MetSy had increased diversity and different compositions of GM in favor of butyrate-producing microbes, leading to an increased insulin sensitivity. These results are in concordance with the study of Le Chatelier et al., in which higher bacterial richness turned out to be a predictive factor for lower body mass index (BMI) and a lower chance of progressing to adiposity-associated co-morbidities [20].

On the other side, decreased microbiota variety, a component of dysbiosis, predisposes to WAT expansion and inflammation. A recently published study showed that diversity of the class *Clostridia*, particularly the richness of the *Vescimonas* genus, correlates with the host’s leanness and metabolic health [36]. Further analysis showed that these bacteria have multiple carbohydrate-active enzymes responsible for butyrate production, besides other benefits on carbohydrate metabolism [36].

GM and its metabolites, particularly SCFAs, modulate appetite and energy intake by interfering with the hunger/satiety-controlling hormones, such as leptin, ghrelin, insulin, GLP-1, and PYY, thus affecting homeostatic and hedonic hunger networks [19,37]. Less-explored succinate, indole, GABA, and different peptide mimetics, including seinolytic peptidase B produced by Escherichia coli, are also made by GM and act as mediators in the gut–brain axis [38,39,40].

Furthermore, GM facilitates energy absorption and modulates the activity of genes related to fat storage and AT expansion via inhibiting the absorption of fatty acids (FA) from the blood into WAT and muscle and promoting the FA oxidation in skeletal muscle and fat cells [33,41]. The AT expansion includes the progressive growth of adipocytes in size (hypertrophy) and number (hyperplasia). These adipocytes become hypoxic and die, releasing inflammatory mediators, such as TNF-α, that recruit macrophages and promote IR [18]. A schematic representation of how gut dysbiosis adversely affects metabolic health is shown in Figure 1.

## 5. Role of the Gut Microbiome in White Adipose Tissue Browning

BAT is a highly active energy-producing tissue located in the interscapular, axillary, cervical, perirenal, and periaortic areas and is entirely different from WAT in both morphological and functional aspects. It is responsible for adaptive (non-shivering) thermogenesis after exposure to a cold environment through β3-adrenergic stimulation. Uncoupling oxidative phosphorylation, mainly in brown adipocytes, via uncoupling protein 1 (UCP1) is the main mechanism, while, accordingly, browning considers the increased expression of UCP1 in WAT depots [42]. The process also includes the activation of PPAR-γ coactivator-1α (PPARGC1A) and thyroid hormone, leading to enhanced mitochondrial biogenesis [18]. Despite numerous research studies, many knowledge gaps and uncertainties persist regarding the relevance of WAT browning in humans. The interest in the subject relies upon the theoretical possibility of enhancing the magnitude of energy expenditure, thereby contributing to weight loss in obese individuals.

Recently, GM emerged as a significant modulator of WAT browning and/or BAT recruitment. After microbiota was transplanted from mice previously exposed to cold to germ-free mice, the process of WAT browning was activated in inguinal and perigonadal depots, leading to fat loss [43]. GM products, primarily SCFAs, can influence host energy expenditure and BAT–WAT balance. When butyrate was administered to obese mice, it activated adaptive thermogenesis and FFA oxidation, increased insulin sensitivity, and reduced adiposity. The upregulation of PPARGC1A in BAT drove the process and WAT-induced AT browning [19,44,45]. These findings in animal models are not fully confirmed in humans.

Apart from SCFAs, GM-derived SBAs, such as deoxycholic acid (DCA) and lithocholic acid (LCA), have been shown to activate thermogenesis in BAT and the browning of WAT [46]. Their effects are achieved by activating the farnesoid X receptor (FXR) and G protein-coupled bile acid receptor-1 (GPBAR-1), also known as Takeda G-protein-coupled receptor 5 (TGR5). When TGR5 is activated, the level of cyclic AMP (cAMP) increases, resulting in mitochondrial biogenesis and changed energy expenditure [47]. Since environmental factors significantly influence the GM composition, when laboratory mice were treated with a mixture of antibiotics, not only was GLP-1 secretion inhibited, browning factors in inguinal WAT were also not expressed [47]. Consequently, alterations in bile acid metabolism found in dysbiosis significantly impact BAT activation, promoting weight gain and obesity.

## 6. Role of the Gut Microbiome in Adipose Tissue Inflammation

As a complex and dynamic organ, the AT modulates glucose and lipid metabolism through the constant secretion of a wide assortment of biologically active molecules with endocrine, paracrine, and autocrine functions. In accordance, adipocyte dysfunction acts as a key driver of metabolic diseases, including IR and lipid overload, as the first step. Investigations aimed at identifying the key pathophysiological mediators of this process ultimately led to the GM as the missing link. An imbalance in the gut microbial consortium can evoke chronic low-grade inflammation in AT, characterized by the phenotypic change and production of pro-inflammatory cytokines.

Bacterial lipopolysaccharide (LPS) is the main endotoxin, which constitutes the outer membrane of the cell wall of Gram-negative bacteria and provides structural integrity to the bacterial cell [41]. In the case of dysbiosis and overgrowth of Gram-negative bacteria, gut epithelial barrier function is altered, and intestinal permeability is increased via the reduction of tight junction proteins. When bacterial death occurs, gut-derived LPS can easily be translocated into the systemic circulation, labeled as metabolic endotoxemia. In conjunction with inflammatory cell receptor CD14, LPS forms an LPS–LBP–CD14 complex. In the next step, the Toll-like receptor 4 (TLR4) signal pathway is included, and the whole complex is presented to Kupffer cells. Based on that interaction, the transcription of proinflammatory genes is activated, and TNF-α, IL-6, and MCP-1 are released, triggering a low-grade inflammation [41].

Studies in animal models showed that when LPS was continuously injected subcutaneously into mice on a normal diet, just a month later, the weight of the mice was significantly increased, and IR was detected [48]. These results shed light on the severity of GM’s impact on metabolic health, even on a regular diet, due to the induction of metabolic endotoxemia. On the contrary, a high-fat diet induces changes in microbiota composition (a decrease in *Bacteroidetes* and an increase in *Firmicutes* species) and the consequent increase in plasma LPS levels [49].

SCFAs play a protective role in strengthening the gut barrier function and preservation of intestinal integrity, while butyrate also exerts potent immunomodulatory and anti-inflammatory properties [50]. The principal mechanisms through which this is being achieved include the activation of G-protein-coupled receptors (GPCRs), inhibition of histone deacetylase (HDAC), suppression of NF-kB activation, inhibition of IFNγ production, and the upregulation of PPAR-γ [51].

## 7. The GM Metabolites Mediate the Gut–Adipose Tissue Crosstalk

Gut microbiome metabolites, including the previously mentioned SCFAs and SBAs, act as molecular bridges in the gut–adipose tissue axis. These metabolites interact with AT’s secretome portfolio, along with some recently introduced players in the crosstalk. Indole propionic acid (IPA), a tryptophan metabolite, is solely GM-produced. Studies have elucidated IPA’s positive impact on glucose metabolism and insulin sensitivity with a reduced risk of developing T2DM and MAFLD [52,53].

In recent years, an additional component, extracellular vesicles (EVs), was added as messengers along the gut–AT axis. EVs are nanovesicles divided into three categories based on their size and biogenesis: exosomes, microvesicles, and apoptotic bodies. These are released by human cells to deliver a wide range of secreted molecules to remote cells and organs within the body [54]. The content of EVs, such as proteins, lipids, nucleic acids, or cellular components, serves as an additional communication tool between distant cells. GM-derived EVs are hypothesized to contribute to T2DM pathogenesis [55]. If the stool EVs from HFD-fed mice are administered to mice fed on a regular diet, it could dampen insulin sensitivity and impair glucose metabolism [55].

Bakke et al. suggested that vitamin D is an additional signaling molecule, modulating the influence of GM on metabolic health. Its actions are accomplished via the vitamin D receptor (VDR). VDR in the liver is activated either by calcitriol, an active form of vitamin D, or by secondary BAs, such as LCA, produced by GM [56].

The communication between immune cells activated by GM products and adipocytes contributes to low-grade inflammation in AT, while cytokines induce IR through direct serine phosphorylation of the insulin receptor substrate 1 or 2 [23]. Molecular crosstalk between GM and host cells activates different metabolic pathways in host cells, resulting in the modulation of host gene expression due to alterations in DNA-binding proteins and epigenetic changes. Enhanced pro-inflammatory and diminished anti-inflammatory states eventually lead to obesity and metabolic dysfunction [18]. Figure 1 illustrates a summary of various mediators related to the gut–adipose tissue axis and the mechanisms involved in the pathogenesis of various components of MetSy and cardiovascular disease.

## 8. The GM Mediates IR and Substrate Metabolism

Initially perceived as a local symbiote involved in digestion, GM is now increasingly recognized for its pivotal role in energy and substrate metabolism in AT, muscle, and the liver, highlighting the possible links between dysbiosis and metabolic diseases [19].

The GM dysbiosis leading to LPS-mediated chronic low-grade inflammation and the demonstration of macrophage infiltration in the AT are the major events in developing obesity-related IR via the inhibition of insulin-stimulated glucose uptake [48,57]. In accordance, circulating LPS levels are significantly higher in obese compared to lean individuals and directly correlate with hemoglobin A1c [58,59]. An activation of the inflammatory cascade promotes adipocyte hypertrophy and fibrosis and the further recruitment of immune cells, aggravating metabolic dysfunction. Exposure to LPS endotoxemia affects insulin receptor signaling via phosphorylation pathways, alters vascular homeostasis, and induces oxidative stress, with increased levels of reactive oxygen species (ROS) [60]. Furthermore, emerging evidence indicates that gut bacteria-derived metabolites upregulate the production of pro-obesity adipokines, such as leptin, resistin, IL-6, MCP-1, and PAI-1, and downregulate the anti-obesity ones, such as adiponectin [60].

A recent international expert forum summarized the state of science on GM and diabetes association and also expressed the need for exploring LPS actions on a specific pathogen-associated molecular pattern (PAMP)–TLR4, together with its role in host–GM crosstalk and the preservation of gut barrier integrity, harboring the potential for new therapeutic approaches [61].

The obesogenic diet, such as HFD, gut–AT axis alteration, and consequent low-grade inflammation, are critical drivers towards developing MetSy, T2DM, and MAFLD. An elevated level of LPS, as a sign of metabolic endotoxemia, highly impacts liver metabolism and causes liver injury due to the activation of hepatic TLR4 signaling, leading to the development and progression of MAFLD [62]. Moreover, choline metabolites produced in the HFD milieu can be further converted into trimethylamine-N-oxide (TMAO) in the liver, which exerts detrimental effects on hepatocytes [63]. GM composition, with a disequilibrium in the *Firmicutes*/*Bacteroidetes* ratio or F/B ratio, directly promotes the progression of liver steatosis [62].

## 9. Modulation of Gut–Adipose Tissue Axis to Improve Metabolic Health

Since the causal link between GM dysbiosis and metabolic disease has been described, a lot of efforts are being made to therapeutically target the gut–adipose tissue axis by modulation of the GM to improve metabolic health, as depicted in Figure 2. Currently, several possible interventions, such as probiotics, prebiotics, and fecal microbiota transplantation (FMT), have shown potential in GM adjustment in combating obesity and related conditions. One of the advantages of this therapy is safety, since no major adverse effects have been reported so far. Still, ongoing and additional research would help determine the most beneficial approach to maintaining metabolic health and restoring a healthy microbiome in both inflammatory and metabolic diseases.

Figure 3 shows a schematic representation of interventions to improve metabolic health through modifying in gut microbiome. 

**Figure 2 cimb-47-00424-f002:**
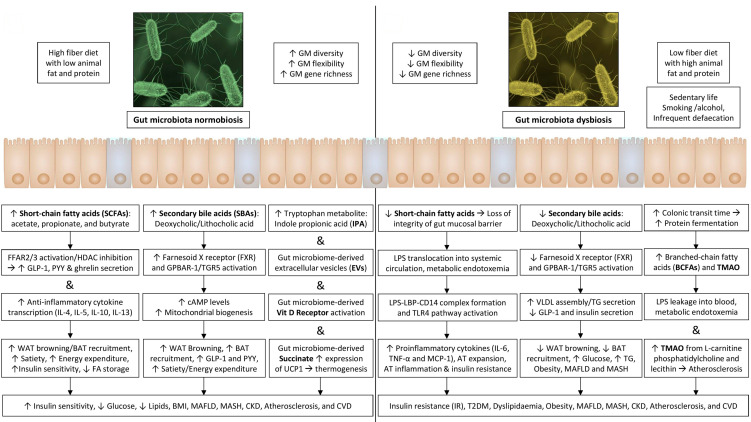
Main mediators in the GM–AT axis and their impact on systemic metabolism.

**Figure 3 cimb-47-00424-f003:**
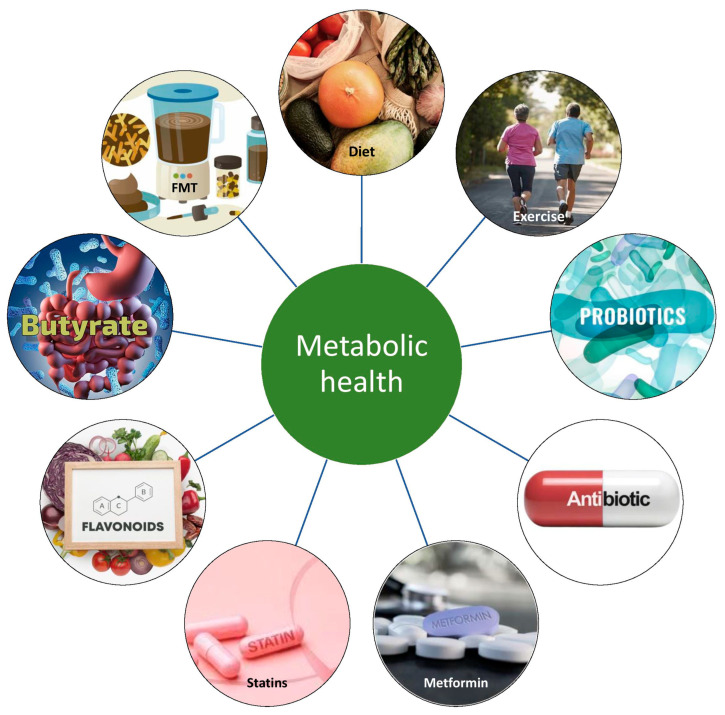
Gut microbiota-based interventions to improve metabolic health.

### 9.1. Nutritional Therapy

The type of diet strongly influences the diversity, composition, and function of GM. Undigested foods serve as a substrate for GM metabolism, shaping its structure and promoting the growth of specific phyla. Studies have particularly explored protein intake, linking its consumption with overall microbial diversity and the abundance of *Bacteroides* and *Alistipes* in their microbial consortium [64,65]. On the other hand, the consumption of a low-fat diet led to the fecal richness of *Bifidobacterium,* with consequent reductions in fasting glucose and total cholesterol compared to baseline, whereas HFD resulted in the reduction of total bacteria and the elevation of LPS [41,66]. The most prominent change in GM composition following HFD consumption is a decrease in *Bacteroidetes* and an increase in *Firmicutes*, alterations that have been linked with obesity and the subsequent development of metabolic diseases [49].

Generally, dietary adjustments, such as a low-fat diet, a diet rich in fiber, or antioxidants, have multiple beneficial aspects for metabolic health. Particularly, a Mediterranean diet, as a diet promoting microbial diversity, has been related to beneficial changes in GM profile characterized by increased growths of *Clostridium leptum*, *Eubacterium rectale*, *Bifidobacteria*, *Bacteroides*, and *Faecalibacterium prausnitzii* species, along with reduced growths of *Firmicutes* and *Blautia* species [67]. Such changes are associated with a favorable AT metabolism and oxidative status and the prevention of systemic inflammation/metabolic disease.

### 9.2. Probiotics, Prebiotics, Symbiotics, and Postbiotics

Currently, the most widely accepted approach for GM modulation involves the use of probiotics and prebiotics. This approach is of particular value in HFD-induced gut dysbiosis, limiting its negative impact on obesity-related disease [49].

Probiotics are preventive or therapeutic preparations that contain active microorganisms and act to restore the composition of GM. Probiotics can potentially manipulate host microbial communities and suppress the growth of pathogenic bacteria by triggering the production of β-defensin and IgA, reinforcing the integrity of the gut epithelial barrier and modulating the immunological response and metabolic health [68]. Prebiotics are non-digestible food components that can precisely adjust the host GM due to selective stimulation of the growth and replication of the beneficial bacteria. Common prebiotics are bifidus factors and oligosaccharides, but many others have been explored as potential targets [42]. A fiber-rich diet also demonstrates prebiotic effects, moving the balance towards the SCFA-producing microbes [13].

Symbiotics are an emerging combination of probiotics and prebiotics that act synergistically to potentiate GM’s favorable effects on health. This approach aims to promote the growth of particular *Bifidobacteria* and *lactobacilli*, influence the release of gastrointestinal hormones such as GLP-1, and reduce obesity-related endotoxemia and IR [41].

Postbiotics, also known as metabiotics or biogenics, are microbial-derived components and molecules that provide certain health benefits to a host. Previously discussed SCFAs and SBAs are the most recognized GM products that influence host metabolic pathways, directly or indirectly, via the endocrine system, immunomodulation, and gut barrier function [13]. Other postbiotics, such as acetate and lactate, produced during intermittent fasting in mice, are thought to bridge the changes in GM and the increased thermogenic capacity, resulting in WAT browning, ameliorating IR, MAFLD, and obesity [42]. Anthocyanins, vanillic acid, and 10-oxo-12(Z)-octadecenoic acid (KetoA) are also postbiotic compounds, currently under investigation for possible benefits, which have already been shown in animal models, with the emerging idea of using them as supplements to combat metabolic diseases [42].

Apart from the extensive research exploring the use of probiotics to modulate the gut–AT axis in obesity and related diseases, including MAFLD, gut dysbiosis is considered to be an aggravating factor in chronic kidney disease [69,70], hypertension [71], psychiatric illness [72], cancer [73,74], and polycystic ovary syndrome [75,76]. Thus, probiotics have emerged as a feasible strategy and an add-on to conventional therapy.

### 9.3. Metformin

Metformin is an effective and safe antidiabetic drug, commonly used as a first-line therapy for T2DM [77]. In treatment-naive patients with T2DM, the metformin-induced GM alterations, registered in 86 strains after 4 months of daily use, contributed to the antidiabetic effects of the drug due to improved glucose tolerance [78]. However, a recently published systematic review of 13 studies provided data analysis of the metformin effects on GM composition, but without consistency across different populations [79].

### 9.4. Supplementation with Bile Acid

Bile acid (BA) exerts pleiotropic effects in humans, particularly in signaling and regulating metabolism, while its interactions with GM play a pivotal role in this process. BA metabolites, such as bile acid sequestrants, stimulate mitochondrial oxidative phosphorylation and promote energy metabolism, modulate BAT levels, and mitigate the development of obesity and T2DM [80]. Beyond that, in experimental and clinical models, cholestyramine improved interactions between GM and BA, facilitating glucose and fat metabolism [81]. In a previously mentioned randomized controlled trial conducted by Wu et al., metformin use also increased concentrations of unconjugated bilirubin (BA), which was associated with lower levels of hemoglobin A1C [78]. Bile acid-activated receptor agonists or antagonists (FXR and TGR5) are evolving as advanced tools to influence several metabolic disorders, and a better understanding of the underlying mechanisms related to GM alterations would provide a significant context for further therapeutic accelerations. One of the investigational compounds is fexaramine, an intestine-restricted farnesoid X receptor (FXR) agonist that induces G protein-coupled bile acid receptor-1 (GPBAR-1) to stimulate GLP-1 secretion but also increases taurolithocholic acid and fibroblast growth factors 15 and 21, resulting in improved insulin and glucose tolerance and promoted WAT browning in mice [46].

### 9.5. The Rational Use of Antibiotics

In the last decades, the caesarean section rate has continuously been on the rise globally with a consequently lower rate of vertical transmission of microbes from the mother, the use of processed food dramatically increased, and the irrational and widespread use of antibiotics became a serious threat, not just for bacterial resistance but also for gut dysbiosis [82]. The frequent and/or prolonged use of wide-spectrum antibiotics can eradicate the specific bacterial phyla, reduce microbial diversity, and promote the overgrowth of antibiotic-resistant microorganisms. Disturbed micro-ecological balance worsens the existing metabolic disease [41].

### 9.6. Fecal Microbiota Transplantation

The idea of fecal microbiota transplantation (FMT) is based on the need for a quick and effective restoration of GM, and there is certain evidence regarding feces use for gastrointestinal disease as early as ancient times in India. Microflora transplantation has already become the standard of care for recurrent or refractory *Clostridium difficile* infections [83,84]. Considering the growing interest in GM’s role in obesity, FMT gained considerable attention, resulting in numerous ongoing clinical studies evaluating its potential role in cardiometabolic medicine.

In a randomized controlled trial, Vrieze et al. examined the effects of FMT from lean donors to male patients with MetSy. Six weeks later, the insulin sensitivity of recipients was significantly improved along with the higher microbial diversity, particularly in butyrate-producing microbiota [35]. Apart from that, FMT is also shown to impact the course of type 1 diabetes mellitus due to the attenuation of the autoimmune process [85].

The stool sample from a healthy individual is considered to contain trillions of beneficial microbes that can ameliorate the recovery of the diseased person [86]. However, the clear limitation of FMT as a therapeutic approach is the heterogeneity and high variability of the donor specimen, characterized by individual GM composition, but also the possibility of short-term and long-term adverse effects. In accordance, the future perspectives of FMT encompass mandatory rigorous donor stool sample screening or the development of a fecal microbiota pill containing a fixed amount of desired microbial phyla.

### 9.7. Therapeutic Options in Development

Current evidence provides new perspectives regarding the therapeutic approach targeting the gut–adipose tissue axis and its connection with metabolic health. Considering its previously described role as a key mediator between GM and the host, it is becoming clear that adequate concentrations of butyrate, a GM-produced SCFA, can help maintain metabolic equilibrium and prevent disease. Administration of dietary supplements of butyrate in mice has been shown to prevent the development of IR and obesity, even on an HFD, due to the effects on the gut–brain neural circuit, promotion of energy expenditure, and unveiling of the mitochondrial function [44,45]. In T2DM rats, butyrate also significantly improved histological alterations of the islet and alleviated β-cell apoptosis [87]. The data derived from human studies showed some benefits, as in the treatment of pediatric obesity, but are still sparse and limited by a small sample size and/or short intervention period, leaving the need for RCTs with longer follow-up to confirm the results [88,89]. Also, the unpleasant organoleptic feature of current butyrate compounds needs to be overcome to improve adherence.

Since the metabolic endotoxemia concept was introduced, LPS was recognized as an inflammatory factor that binds to CD14 and acts as a vector for the development of IR and obesity induced by HFD [90]. Lowering plasma LPS concentrations could potentially interrupt the chain of the development of metabolic diseases. A possible beneficial effect of flavonoids, the plant components, in the induction of GM changes in obesity models has also been examined. Quercetin, resveratrol, fermented green tea extract, and polyphenol-rich cranberry extract need to be evaluated in studies to develop effective interventions [49,91]. Statin therapy was found to be associated with an increase in *Faecalibacterium* and a decrease in *Bacteroides* in obese individuals taking statins, and this can be associated with a lowered risk of inflammatory bowel diseases [92]. Treatments that target TMAO, the GM metabolite, have been tested in CKD and CVD [93,94,95].

### 9.8. Effects of Bariatric Surgery on GM and the Implications on Metabolic Health

Bariatric surgery has emerged as the most effective intervention for sustained weight loss and comorbidity improvement. Alterations in gut microbiota significantly contribute to metabolic improvement, such as enhanced glucose control and lipid metabolism in patients following bariatric surgery. A recent systematic review looking at the impact of bariatric surgery on gut microbiota has shown an increased microbial diversity, reduced *Firmicutes,* and elevated beneficial bacteria, such as *Akkermansia muciniphila* and short-chain fatty acid-producing bacteria. The microbiota changes were correlated with significant improvements in weight loss, insulin sensitivity, and lipid profiles [96]. However, the long-term sustainability of these microbial changes remains unclear.

Sleeve gastrectomy and gastric bypass are commonly performed surgical procedures. Both have similar outcomes and result in a healthier patient, improving MetSy and obesity-related complications. The patient’s inflammatory state due to MetSy improves after surgery [97]. A recent study reported that GM alterations induced by Roux-en-Y gastric bypass result in glucose-lowering by enhancing intestinal glucose excretion [98].

The GM has been emerging as a promising source of potential therapeutic targets for modulating neuroinflammation and food reward alteration. It is worth exploring GM-targeted interventions in the development of novel treatment approaches towards eating disorders [99]. A recent systematic review reports an increasing trend in research and scientific contributions on the impact of bariatric surgery on GM, the biomechanisms involved, its role in the weight loss process, and the development of treatment modalities based on GM [100].

In summary, bariatric surgery has a multifaceted role in reducing caloric intake, GI hormonal secretion, changes in gut microbiome, and immunomodulatory mechanisms.

## 10. Summary and Conclusions

In the past two decades, overwhelming evidence has accumulated regarding the link between the gut–AT axis and metabolic health/disease. The underlying mechanisms of these gut–AT interactions are of considerable interest in current scientific research. The genetic sequencing methods allowed researchers to identify the specific phyla among GM and link them to the metabolic alterations. However, the GM is a vivid and diverse consortium across the lifespan of an individual, and it becomes increasingly clear that it is not appropriate to perceive it as “good” or “bad” but rather to explore the full spectrum of possibilities.

Advances in knowledge concerning biological pathways provide a base for the nutritional prevention of gut dysbiosis and open a door to novel therapeutic strategies targeting the complex interplay between GM and the host in the preservation of metabolic health or the restoration of disturbed metabolic functions. Even with a lot of gaps in current evidence, mostly because GM is being hindered by a huge rate of diversity, research in this field continues to evolve, encouraging an interdisciplinary merging across microbiology, endocrinology, immunology, nutrition, and neuroscience. Future efforts should focus on the design of prospective randomized clinical trials to validate data generated from animal models.

The flow diagram below (Figure 4) shows potential therapeutic strategies for improving a gut microbiome profile for better health.

## Figures and Tables

**Figure 1 cimb-47-00424-f001:**
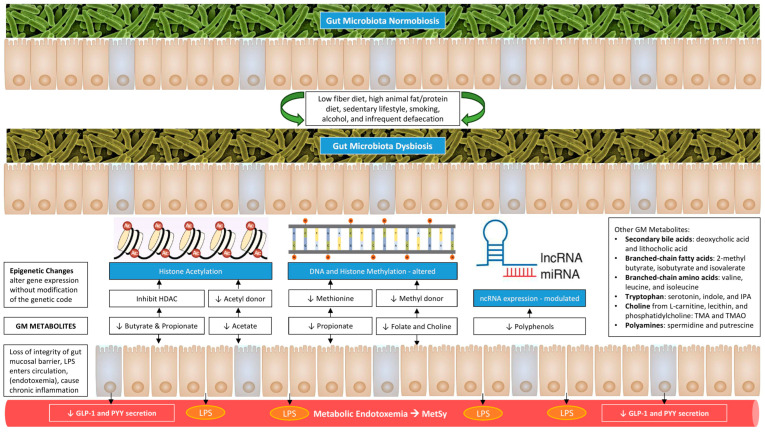
Gut dysbiosis and its adverse consequences on metabolic health.

**Figure 4 cimb-47-00424-f004:**
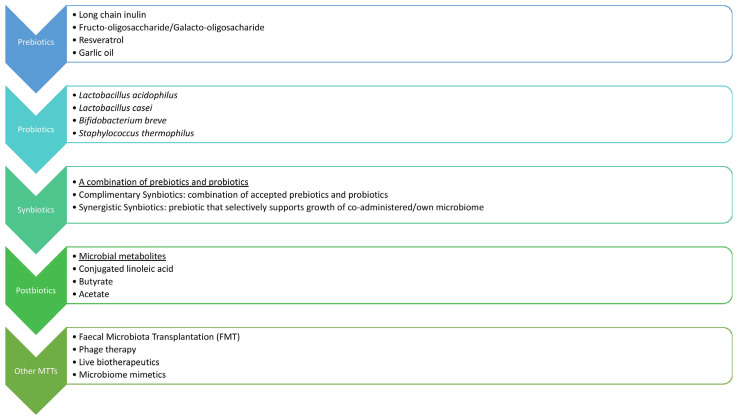
Flow chart showing interventions to improve metabolic health through manipulation of gut-adipose tissue axis.

## Data Availability

Data sharing is not applicable.

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
