# Peer review of "Gut–Adipose Tissue Axis and Metabolic Health"

_cimb, 2025, doi:10.3390/cimb47060424_

Round 1
Reviewer 1 Report
Comments and Suggestions for Authors
cimb-3644559-peer-review-v1
The paper is an interesting review, providing information about Gut-Adipose Tissue Axis and Metabolic Health, and in my opinion authors have combined significant amount of information. However, some parts can be extended, since some of the topics deserve a bit more attention. One general criticism of the manuscript is that in my opinion authors have been a bit negligent in the preparation and formatting the manuscript. In several parts the manuscript is missing attention from the authors. Please, maybe you can ask some of your more experienced colleagues to proof read the manuscript and to adjust some of the parts.
Please, present the title according to the Instructions for authors.
In the entire manuscript, reference need to be part of the sentence, and full stop to be after the reference; example Ln31: "cytokines [1]. "
Ln103: Please, can it be more specific?
Microbial names needs to be in italics (see Ln115 and etc). Regarding use of genus name Lactobacillus, since reclassification of the genus Lactobacillus into the 23 new genera in 2020, it was suggested, when referring to the term Lactobacillus in sense of meaning before 2020, to use English word " lactobacilli", written without italics and non-capital L.
Maybe authors will consider extending section 4 with some more examples. This is an important topic, and in my opinion deserve a bit more attention. In fact, all sections from 3 to 8 deserve to be extended.
Please, provide information about title of figure 1
Please, provide a description for Figure 2.
Maybe under topic 9.2. will be appropriate if authors will extend text with some information about postbiotics.
Ln458: Akkermansia Munciphilia, needs to be Akkermansia munciphilia, and to in italics. Please, check the entire manuscript for similar adjustments for negligence to be corrected.
Reference needs to be formatted according to the instructions form the Publisher and The Journal.
Author Response
Comment 1:
Please, present the title according to the Instructions for authors.
Response 1:
According to the instructions for authors, a running title was deleted.
Comment 2:
In the entire manuscript, reference need to be part of the sentence, and full stop to be after the reference; example Ln31: "cytokines [1]. "
Response 2:
References in text were changed to be the integral part of the sentence.
Comment 3:
Ln103: Please, can it be more specific?
Response 3:
Accurate data mentioned, as suggested
Comment 4:
Microbial names need to be in italics (see Ln115 and etc). Regarding use of genus name Lactobacillus, since reclassification of the genus Lactobacillus into the 23 new genera in 2020, it was suggested, when referring to the term Lactobacillus in sense of meaning before 2020, to use English word " lactobacilli", written without italics and non-capital L.
Response 4:
Microbial names modified, in italics, ‘’Lactobacillus’’ changed to ‘‘lactobacilli’’
Comment 5:
Maybe authors will consider extending section 4 with some more examples. This is an important topic, and in my opinion deserve a bit more attention. In fact, all sections from 3 to 8 deserve to be extended.
Response 5:
Section 3 and 4 are extended with additional examples
Comment 6:
Please, provide information about title of figure 1
Response 6:
A description about the figure 1 is provided
Comment 7:
Please, provide a description for Figure 2.
Response 7:
A description on Figure 2 is provided
Comment 8:
It will be appropriate if authors will extend text with some information about postbiotics.
Response 8:
The paragraph on postbiotics is added to section 9.2
Comment 9:
Ln458: Akkermansia Munciphilia, needs to be Akkermansia munciphilia, and to in italics. Please, check the entire manuscript for similar adjustments for negligence to be corrected.
Response 9:
The entire manuscript checked for accuracy
Comment 10:
Reference needs to be formatted according to the instructions form the Publisher and The Journal.
Response 10:
References are formatted
Reviewer 2 Report
Comments and Suggestions for Authors
This work aimed to discuss the genetic, epigenetic, and molecular aspects of the gut-adipose tissue axis and its modulation to improve metabolic health. However, there is a lack of figures and tables for the research of this method, and the content of the manuscript is boring to read. The author needs to organize the key contents into corresponding figures and tables to improve the quality of the manuscript. In addition, the framework content of the manuscript needs to be adjusted. There are too many titles, and the review content is not concentrated enough. It is suggested to summarize it according to metabolites and regulatory methods. The manuscript should be double-checked before it can be considered for publication by CIMB.
Author Response
This work aimed to discuss the genetic, epigenetic, and molecular aspects of the gut-adipose tissue axis and its modulation to improve metabolic health.
Comment 1:
There is a lack of figures and tables for the research of this method, and the content of the manuscript is boring to read. The author needs to organize the key contents into corresponding figures and tables to improve the quality of the manuscript.
Response 1
Key contents are now organised in two additional figures
Comment 2:
The framework content of the manuscript needs to be adjusted. There are too many titles, and the review content is not concentrated enough. It is suggested to summarize it according to metabolites and regulatory methods.
Response 2:
The manuscript underwent extensive revision. Though it is suggested to adjust the framework according to metabolites and regulatory methods, we feel that, our format would be helpful for the new readers of this topic who are unfamiliar with the complexities.
Comment 3:
The manuscript should be double-checked before it can be considered for publication by CIMB.
Response 3:
The manuscript is checked multiple times to ensure accuracy.
In addition we have made appropriate language corrections to improve the quality of the manuscript as suggested.
Thanks for the time and patience to review the paper.
We have addressed the issues in the revision as suggested which improved the quality of paper substantially.
